# Driving Waveform Design with Rising Gradient and Sawtooth Wave of Electrowetting Displays for Ultra-Low Power Consumption

**DOI:** 10.3390/mi11020145

**Published:** 2020-01-28

**Authors:** Wei Li, Li Wang, Taiyuan Zhang, Shufa Lai, Linwei Liu, Wenyao He, Guofu Zhou, Zichuan Yi

**Affiliations:** 1Guangdong Provincial Key Laboratory of Optical Information Materials and Technology & Institute of Electronic Paper Displays, South China Academy of Advanced Optoelectronics, South China Normal University, Guangzhou 510006, China; wei.li@guohua-oet.com (W.L.); taiyuan.zhang@guohua-oet.com (T.Z.); shufa.lai@guohua-oet.com (S.L.); linwei.liu@guohua-oet.com (L.L.); guofu.zhou@m.scnu.edu.cn (G.Z.); 2Shenzhen Guohua Optoelectronics Tech. Co., Ltd., Shenzhen 518110, China; yizichuan@163.com; 3Academy of Shenzhen Guohua Optoelectronics, Shenzhen 518110, China; 4College of Electron and Information, University of Electronic Science and Technology of China, Zhongshan Institute, Zhongshan 528402, China

**Keywords:** driving waveform, power consumption, electrowetting display, aperture ratio, response speed

## Abstract

As a kind of paper-like display technology, power consumption is a very important index for electrowetting displays (EWDs). In this paper, the influence of driving waveforms on power consumption of the EWDs is analyzed, and a driving waveform with rising gradient and sawtooth wave is designed to reduce the power consumption. There are three stages in the proposed driving waveform. In the initial stage, the driving voltage is raised linearly from the threshold to the maximum value to reduce the invalid power consumption. At the same time, the oil breakup can be prohibited. And then, a section of sawtooth wave is added for suppressing oil backflow. Finally, there is a section of resetting wave to eliminate the influence of charge leakage. Experimental results show that the power consumption of the ultra-low power driving waveform is 1.85 mW, which is about 38.13% lower than that of the conventional used square wave (2.99 mW), when the aperture ratio is 65%.

## 1. Introduction

The electrowetting display (EWD) is a kind of reflective display technology, which has the advantages of video updating rate, high contrast ratio and low power consumption. Its working mechanism is to manipulate the interfacial tension of the polar liquid in a pixel by applying electric field, to achieve the control of optical switch and gray scale display [1]. The EWD technology has been proposed by G. Beni in 1981 at the first time [2]. In 2003, Robert A. Hayes realized high quality EWD [1], and then, a three-color EWD with vertical stack structure was proposed in 2010 [3].

In order to improve the display quality of EWDs, interface materials, pixel structures, fluidic motion mechanism and the relationship between electro-optical performances and applied voltage have been studied [4,5,6,7]. At the same time, the influence of driving waveforms on the performance of EWDs have been also studied. A decoupling driving waveform has been proposed to diminish the induced voltage stress [8]. However, the response time of the EWD is as long as 30 ms which cannot be used to display video. Then, a driving waveform was proposed for realizing the display of multi gray scales according to the characteristics of EWDs, square wave with high frequency is used for reducing flicker, a reset process is added to eliminate the influence of charge leakage [9]. However, the high frequency will lead to high power consumption. At the same time, a method was proposed to reduce oil breakup by adding a voltage rising gradient in the driving waveform [10]. However, the rising gradient can increase the response time of EWDs. Recently, a dynamic and asymmetric driving waveform was proposed to achieve more gray levels with limited voltage levels [11]. Although the power consumption has been decreased by restricting the maximum voltage of the dynamic and asymmetric driving waveform, the maximum aperture ratio cannot be achieved due to the limitation of the maximum voltage.

In this paper, according to EWD’s equivalent circuit model, the power consumption of conventional driving waveforms is calculated and measured. Then, the relationship between the driving waveform and the power consumption is studied based on the experimental data. A driving waveform for ultra-low power consumption is proposed by reducing the driving voltage and shortening the effective driving time in each frame. With the same display effect, the proposed driving waveform can achieve lower power consumption than conventional waveforms.

## 2. Principle of EWDs

Gray scale is realized in EWDs by applying an external voltage to control the movement of colored oil. Its essence is an optical switch, which has excellent gray scale display characteristics [12]. As shown in Figure 1a, the colored oil in the pixel spreads naturally and covers the whole pixel when no external voltage is applied, and the color of the oil is displayed; the colored oil is pushed to a corner in the pixel when the external voltage is applied, and the color of the substrate is displayed, as shown in Figure 1b. Figure 1c is the top view of oil spreading. Figure 1d is the top view of oil contracting. The different degrees of oil contraction represent different optical states, which are characterized by the aperture ratio. The aperture ratio is the proportion of the opening area in the whole pixel. The formula is defined as follows [5]:(1)$$W_{A}\left(V\right)=\left(1-\frac{S_{oil}\left(V\right)}{S_{pix}}\right)\times 100\%$$

In Equation (1), $W_{A}\left(V\right)$ represents the aperture ratio, $S_{oil}\left(V\right)$ and $S_{pix}$ represent the surface area of oil in a single pixel and the surface area of the whole pixel respectively, V represents the voltage applied to the EWD, and the area of pixel wall is ignored in calculating the aperture ratio. The pixel wall is a transparent grid structure which can divide the EWD into several pixels.

The pixel structure of the EWD is shown in Figure 2a, which is mainly composed of glass substrate, ITO (Indium Tin Oxides) guide electrode, hydrophobic insulation layer, pixel wall, color oil, and polar liquid [13]. The three-dimensional structure of an EWD pixel is shown in Figure 2b.

In order to display different gray scales in EWDs, it is necessary to apply a voltage sequence for driving the pixel, the voltage sequence is called as driving waveform [14]. The driving waveform is designed as PWM (pulse width modulation) or AM (amplitude modulation) for displaying different gray scales accurately in EWDs [15]. The driving waveform affects the display effect of the EWD seriously. Figure 3 shows the conventional driving waveforms [9,10].

## 3. Driving Waveform Design for Low Power Consumption

### 3.1. Power Consumption Analysis

As a capacitive device, the equivalent circuit model of an EWD pixel can be defined as a R-C (Resistance-Capacitance) circuit, as shown in Figure 4 [16]. In Figure 4a, $C_{p}$ represents the capacitance of the pixel unit, $R_{p}$ represents the resistance of the pixel unit, $R_{E}$ represents the external resistance, and the E is the driving voltage. Each pixel unit can be represented by a resistor connected in parallel with a capacitor. All pixel units are connected in series or parallel to form a whole. And then, the whole units are connected in series with an external resistance. This is the complete equivalent circuit model of the EWD. Figure 4b is the simplified equivalent circuit model of an EWD pixel. $C_{S}$ represents the capacitance of an EWD pixel, $R_{S}$ represents the resistance of an EWD pixel.

So, the power consumption of the simplified equivalent circuit can be shown as Equation (2).
(2)$$P=\frac{W}{T}=\frac{\frac{U^{2}}{R}t}{T}=\frac{U^{2}t}{TR}$$

In Equation (2), P represents the power consumption of the EWD, W represents the electric energy consumed in a unit time T, U represents the driving voltage, t represents the effective driving time in a unit time T, and R represents the impedance of the EWD. Because of the following theoretical analysis and experimental verification, the impedance of the EWD used in this paper is basically unchanged at the frequency of 60 Hz.

According to the circuit model, an EWD pixel is an R-C series parallel circuit, as shown in Figure 5. There are two break frequencies f1 and f2 in R-C series parallel circuit [17,18], as shown in Equations (3) and (4).
(3)$$f1=\frac{1}{2\pi R_{2}C_{E}}$$
(4)$$f2=\frac{1}{2\pi C_{E}\left(\frac{R_{1}R_{2}}{R_{1}+R_{2}}\right)}$$

In the Equations (3) and (4), f1 and f2 are the break frequencies, $C_{E}$ is the capacitance, $R_{1}$ and $R_{2}$ are series resistance and parallel resistance respectively.

When the frequency is lower than f1, $C_{E}$ is equivalent to an open circuit, and the total impedance of the circuit is *R*_1_ + *R*_2_. When the frequency is higher than f2, $C_{E}$ is equivalent as a short circuit, and the total impedance of the circuit is $R_{1}$. When the frequency is higher than f1 and lower than f2, the total impedance of the circuit can be changed between $R_{1}+R_{2}$ and $R_{1}$.

It can be seen from the Figure 5b that the impedance remains unchanged at a fixed frequency. The impedance of the EWD used in this paper is 170.43 kΩ measured by the TH2828 Precision LCR Meter (Tonghui, Changzhou, China) at the frequency of 60 Hz.

According to the above analysis, the driving power of the EWD is mainly determined by the driving voltage U and the effective driving time t, which can be expressed as the Equation (5).
(5)$$P=\frac{1}{TR}\int_{0}^{T}U^{2}\left(t\right)dt$$
where U(t) is the voltage of the driving waveform at time t.

### 3.2. Power Consumption Analysis of Conventional Driving Waveforms

For conventional driving waveforms, such as square wave and trapezoidal wave, their expressions and power consumption expressions can be derived.

Square wave is shown in Equation (6). $C_{1}$ is the voltage constant from 0 to $\frac{1}{2}T$, and $C_{2}$ is the voltage constant from $\frac{1}{2}T$ to T.
(6)$$U\left(t\right)=\{\begin{array}{cc} C_{1}, & 0\leq t<\frac{1}{2}T\\ C_{2}, & \frac{1}{2}T\leq t<T \end{array}$$

Then, the power consumption expression of Square Wave can be obtained, as shown in Equation (7).
(7)$$P=\frac{1}{TR}\int_{0}^{\frac{1}{2}T}C_{1}{}^{2}dt+\frac{1}{TR}\int_{\frac{1}{2}T}^{T}C_{2}{}^{2}dt$$

Trapezoidal wave is shown in Equation (8). k is the slope between 0 and $\frac{1}{4}T$, $C_{1}$ is the voltage constant from $\frac{1}{4}T$ to $\frac{1}{2}T$, and $C_{2}$ is the voltage constant from $\frac{1}{2}T$ to T.
(8)$$U\left(t\right)=\{\begin{array}{ll} kt, & 0\leq t<\frac{1}{4}T\\ C_{1}, & \frac{1}{4}T\leq t<\frac{1}{2}T\\ C_{2}, & \frac{1}{2}T\leq t<T \end{array}$$

Then, the power consumption expression of trapezoidal wave can be obtained, as shown in Equation (9).
(9)$$P=\frac{1}{TR}\int_{0}^{\frac{1}{4}T}{\left(kt\right)}^{2}dt+\frac{1}{TR}\int_{\frac{1}{4}T}^{\frac{1}{2}T}C_{1}{}^{2}dt+\frac{1}{TR}\int_{\frac{1}{2}T}^{T}C_{2}{}^{2}dt$$

Sawtooth wave is shown in Equation (10). k is the slope between 0 and $\frac{1}{2}T$, $C_{1}$ is the voltage constant from $\frac{1}{2}T$ to T.
(10)$$U\left(t\right)=\{\begin{array}{ll} kt, & 0\leq t<\frac{1}{2}T\\ C_{1}, & \frac{1}{2}T\leq t<T \end{array}$$

Then, the power consumption expression of Sawtooth Wave can be obtained, as shown in Equation (11).
(11)$$P=\frac{1}{TR}\int_{0}^{\frac{1}{2}T}{\left(kt\right)}^{2}dt+\frac{1}{TR}\int_{\frac{1}{2}T}^{T}C_{1}{}^{2}dt$$

### 3.3. Driving Waveform Design

The driving waveform for low power consumption is a sequence of voltages. In order to design a perfect driving waveform, not only the frequency, amplitude and duty cycle of the waveform should be considered, but also the inherent defects of the EWD should be avoided by appropriate optimization and adjustment. In the proposed driving waveform, the frequency is set as 60 Hz [19], the voltage can be changed between 0 and 35 V, and the duty is set as 50%. Based on the square wave, the proposed driving waveform is shown in Figure 6a, which consists of three stages.

The first stage is a linear rising process. The driving voltage is changed from 10 V to 35 V in 5 ms with a linear gradient. The 10 V is the threshold voltage for driving the oil. The oil is in a balance state of natural spreading when no voltage is applied. In order to break this balance state, the threshold voltage needs to be applied. As shown in Figure 6b, the static balance can be broken when the driving voltage is improved to 10 V.

In addition, the rising gradient from 10 V to 35 V is added for reducing the effect of oil rupture on the aperture ratio [10]. At the same time, it can reduce the duration of the driving cycle. The aperture ratio of different rising gradients is shown in Figure 6c. The rising duration is designed as 5ms for reducing power consumption.

The second stage is a periodic change process. The driving voltage is changed between 31 V and 35 V in 3 ms, which is a sawtooth wave. The sawtooth wave can reduce the driving voltage for saving the power consumption and inhibit the oil backflow to a certain extent. The oil will backflow due to charge trapping when the same voltage is applied continuously [8]. The 31 V is the minimum driving voltage of the sawtooth wave. Although the lower driving voltage can reduce more power consumption, the lower driving voltage can reduce the aperture ratio. In order to obtain higher aperture ratio and lower power consumption, 31 V is selected as the minimum driving voltage for the sawtooth wave. The relationship between minimum driving voltage of the sawtooth wave and aperture ratio is shown in Figure 6d, and a higher aperture ratio can be obtained at 31 V. In order to achieve a higher aperture ratio, a higher voltage is needed, but the oil will backflow when a constant voltage is applied to the EWD. And the sawtooth wave can inhibit the oil backflow to a certain extend. So, the aperture ratio will fluctuate locally under the driving of constant voltage and sawtooth wave. Further, for different EWDs, the minimum driving voltages of the sawtooth wave are different, which can be obtained by the same experimental method.

The third stage is a resetting process. The driving voltage is held at 0 V for 8 ms to eliminate the influence of charge leakage [9].

According to the proposed driving waveform, its expression can be expressed as Equation (12). $k_{1}$ is the slop, $b_{1}$ is the intercept between 0 and $\frac{5}{16}T$, $k_{2}$ is the slop, $b_{2}$ is the intercept between $\frac{5}{16}T$ and $\frac{1}{2}T$, C represents the voltage constant from $\frac{1}{2}T$ to T.
(12)$$U\left(t\right)=\{\begin{array}{cc} k_{1}t+b_{1}, & 0\leq t<\frac{5}{16}T\\ k_{2}\left(t-a\right)+b_{2}, & \frac{5}{16}T\leq t<\frac{1}{2}T\\ C, & \frac{1}{2}T\leq t<T \end{array}$$

Then, the power consumption expression of the proposed driving waveform can be expressed as Equation (13).
(13)$$P=\frac{1}{R}\int_{0}^{\frac{5}{16}T}{\left(k_{1}t+b_{1}\right)}^{2}dt+\frac{1}{R}\int_{\frac{5}{16}T}^{\frac{1}{2}T}{\left[k_{2}\left(t-a\right)+b_{2}\right]}^{2}dt+\frac{1}{R}\int_{\frac{1}{2}T}^{T}C^{2}dt$$

## 4. Experimental Results and Discussion

For the sake of evaluating the performance of the proposed driving waveform, an experimental platform is developed to measure the aperture ratio and the power consumption of EWDs, as shown in Figure 7. This experimental platform includes a driving system, a measuring system, and an EWD panel. The driving system, which is used to input the driving waveform for driving the EWD, consists of a computer, a function generator, and a high-voltage amplifier. The testing system is used to measure and record test results, including a microscope, a high-speed camera, a power meter, and a computer, and the EWD is the measured object. Its parameters are shown in the Table 1.

The experimental process can be divided into two steps. Firstly, the EWD is driven by the driving system. The driving waveform is edited by a computer with ArbexPress software, and it is sent to function generator by serial communication, and then, the driving voltage in the driving waveform can be output when it is amplified by the high-voltage amplifier for driving the EWD. Secondly, the experimental data is measured by the testing system. The high-speed camera captures the display state of the EWD in real time by the microscope and transmits testing data to the computer for calculating the aperture ratio. Meanwhile, the power meter measures and records the power consumption of the EWD panel in real time. During the experiment, the temperature is kept constant to avoid the influence of the external environment. 

The aperture ratio and power consumption of the EWD with different driving waveforms are measured by the above-mentioned experimental platform. The measurement results are shown in Figure 8.

As can be seen from Figure 8, with the same aperture ratio, the power consumption of the proposed driving waveform is lower than that of the conventional driving waveforms for the EWD. As shown in Figure 9, the aperture ratio of the proposed driving waveform is larger than that of conventional driving waveforms when the power consumption is 2 mW.

More experiments are designed to further illustrate the characteristics of the proposed driving waveform. As shown in Figure 10a, with the increase of the aperture ratio, the influence of the voltage on it decreases gradually. And as can be seen from the Figure 10b, the driving frequency also have an effect on the aperture ratio. The aperture ratio increases with the increase of driving frequency, but there will be local fluctuation. Figure 10c shows the relationship between aperture ratio and power.

Similarly, the structural parameters of the EWD can affect its power consumption and aperture ratio. However, the shape of the proposed driving waveform cannot be affected, nor will it affect the result in that the proposed driving waveform has lower power consumption than the conventional square wave under the same aperture ratio.

## 5. Conclusions

In this paper, the power consumption of the driving waveform for EWDs is analyzed, and the relationship between the driving waveform and the power consumption is obtained. Then, a new driving waveform with a rising gradient and a sawtooth wave is proposed. The rising gradient can prohibit the oil breakup effectively. The section of sawtooth wave can inhibit the backflow of the oil. Experimental results show that the proposed driving waveform has a lower power consumption and higher aperture ratio than conventional waveforms. When the aperture ratio is 65%, the power consumption of the proposed driving waveform is about 38.13% lower than that of the conventionally used square wave. In addition, the proposed driving waveform can provide a new way for the optimization of the EWD driving system.

## Figures and Tables

**Figure 1 micromachines-11-00145-f001:**
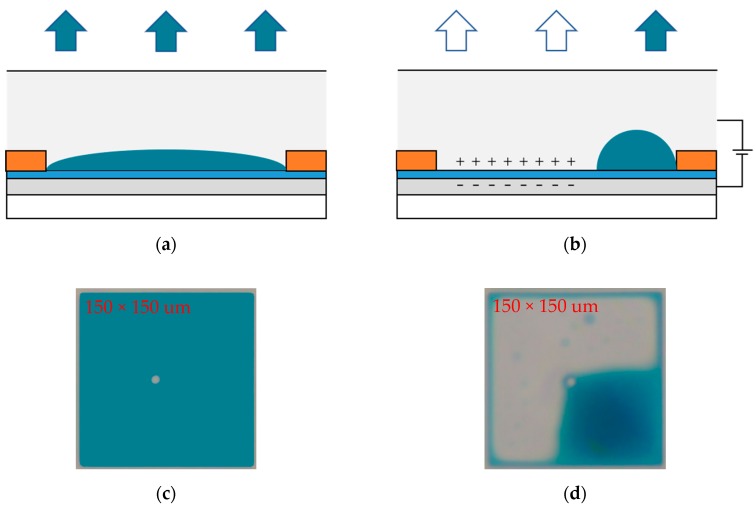
Principle of the electrowetting display (EWD). (**a**) Without applied voltage, the color of oil is displayed in a pixel. (**b**) With applied voltage, the color of substrate is displayed. (**c**) The top view when the pixel is turned off. (**d**) The top view when the pixel is turned on.

**Figure 2 micromachines-11-00145-f002:**
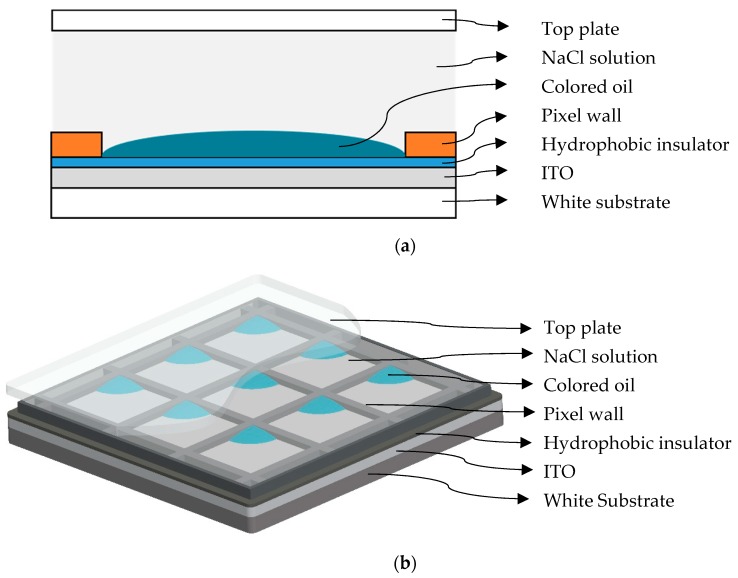
Structure of the EWD. (**a**) Pixel structure of the EWD. (**b**) Three-dimensional structure of an EWD panel.

**Figure 3 micromachines-11-00145-f003:**
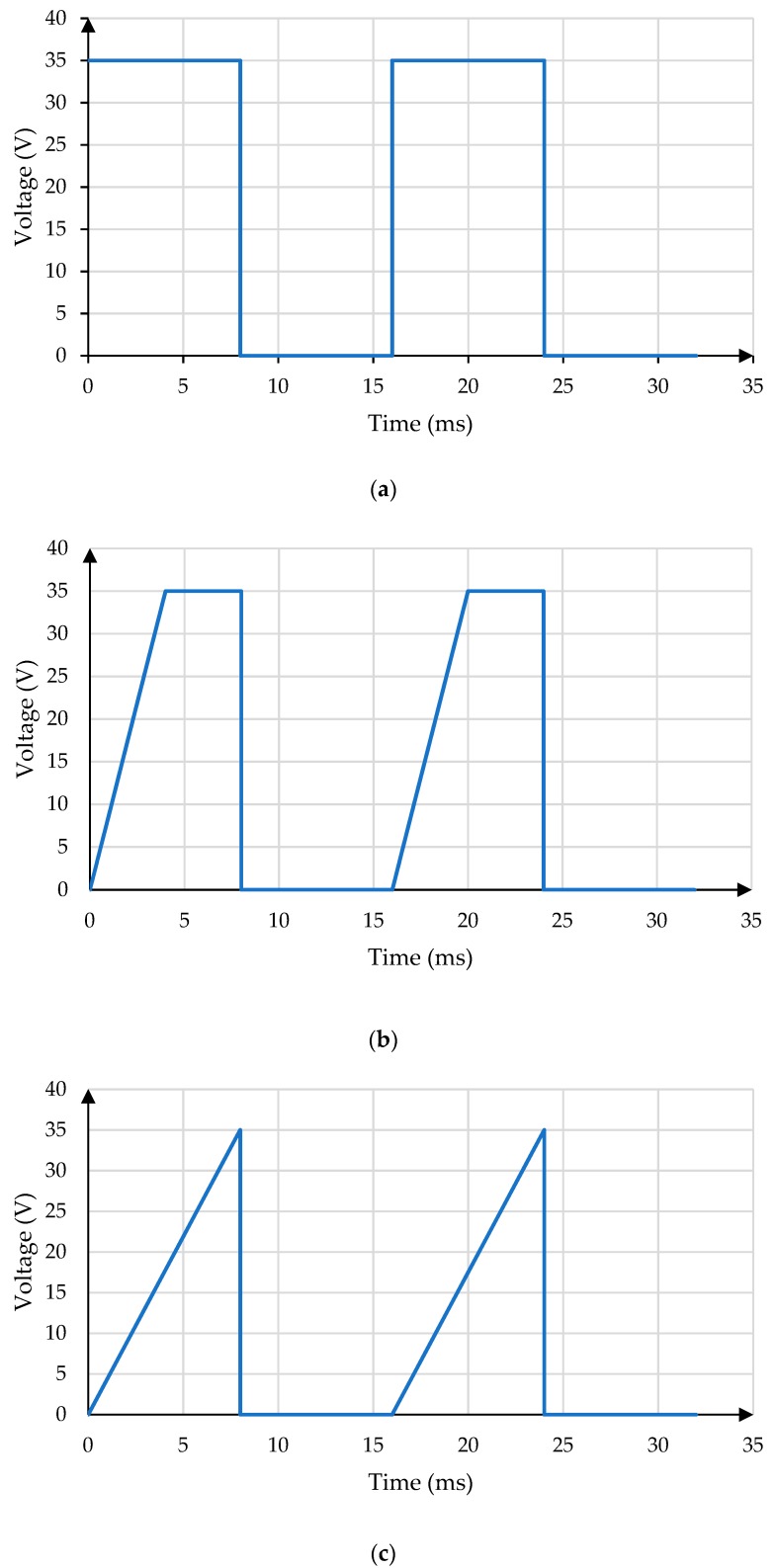
Conventional driving waveforms. (**a**) Square Wave. (**b**) Trapezoidal Wave. (**c**) Sawtooth Wave.

**Figure 4 micromachines-11-00145-f004:**
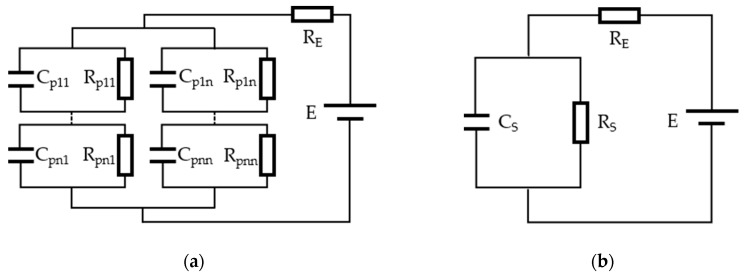
The equivalent circuit models of an EWD pixel. (**a**) Complete equivalent circuit model. (**b**) Simplified equivalent circuit model.

**Figure 5 micromachines-11-00145-f005:**
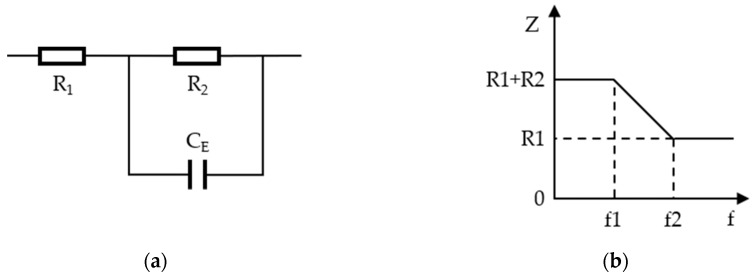
The impedance-frequency characteristics of the R-C series-parallel circuit. (**a**) R-C series-parallel circuit. (**b**) The relationship between the impedance and the break frequency.

**Figure 6 micromachines-11-00145-f006:**
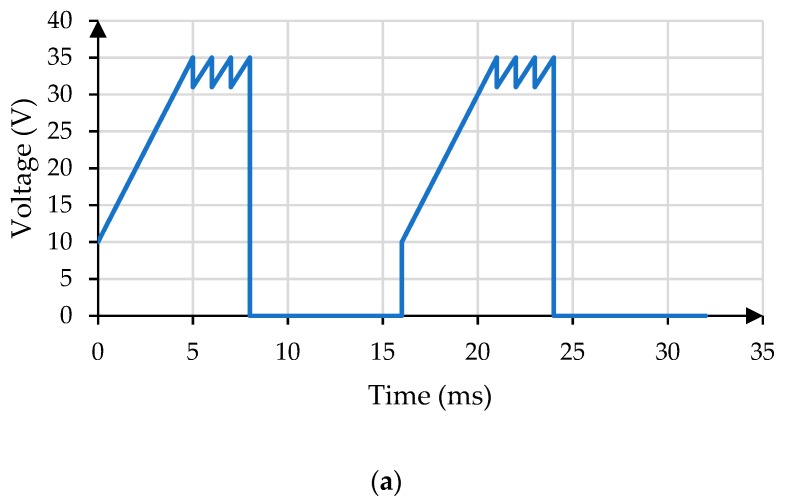
Design of the proposed driving waveform. (**a**) The proposed driving waveform for low power consumption. (**b**) The relationship between the voltage and the aperture ratio. (**c**) The relationship between the rising duration and the aperture ratio. (**d**) The relationship between minimum driving voltage and Aperture Ratio in the sawtooth wave.

**Figure 7 micromachines-11-00145-f007:**
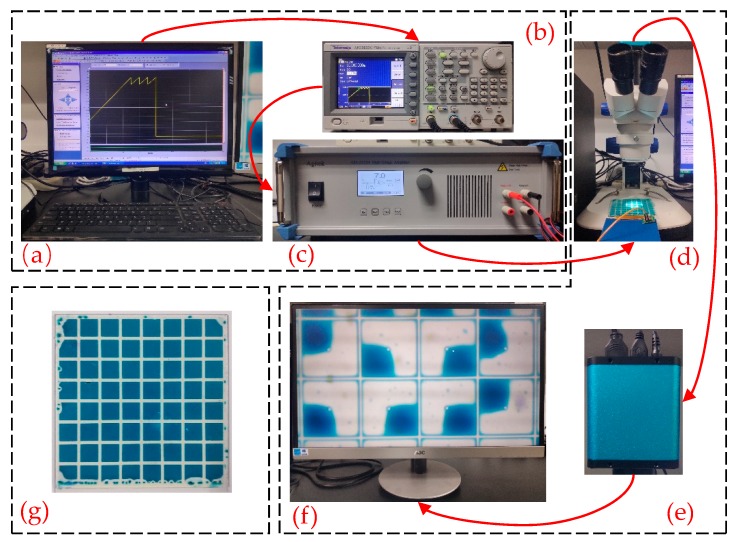
Experimental platform of the driving waveform for EWDs. (**a**) A computer for driving system. (**b**) Arbitrary function generator ATA-2022H (Agitek, Xi’an, China). (**c**) AFG-3052C (Agitek, Xi’an, China) high voltage amplifier. (**d**) Microscope. (**e**) High speed camera. (**f**) PC for measuring system. (**g**) An EWD panel.

**Figure 8 micromachines-11-00145-f008:**
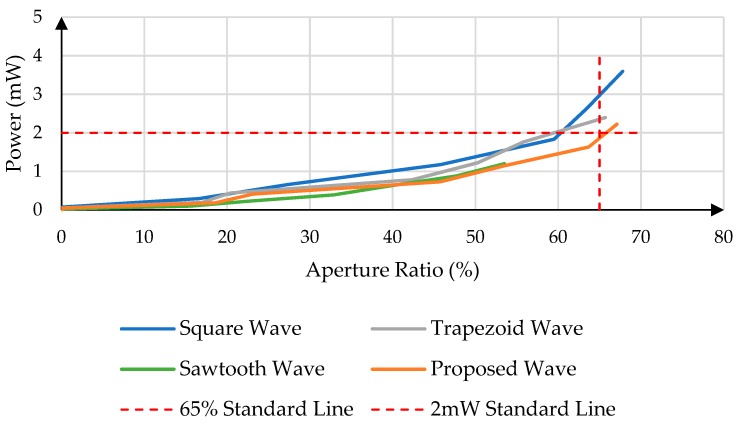
The relationship between Aperture Ratio and Power Consumption.

**Figure 9 micromachines-11-00145-f009:**
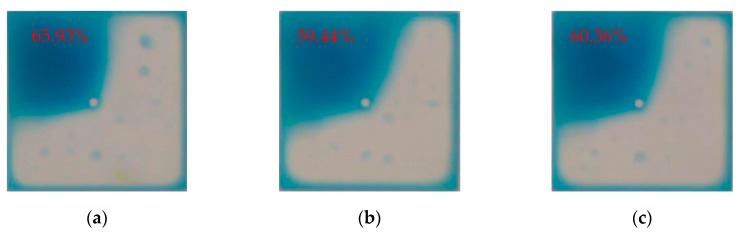
Aperture ratio with different driving waveforms under the same power consumption. (**a**) The proposed driving waveform. (**b**) Trapezoid driving waveform. (**c**) Square driving waveform.

**Figure 10 micromachines-11-00145-f010:**
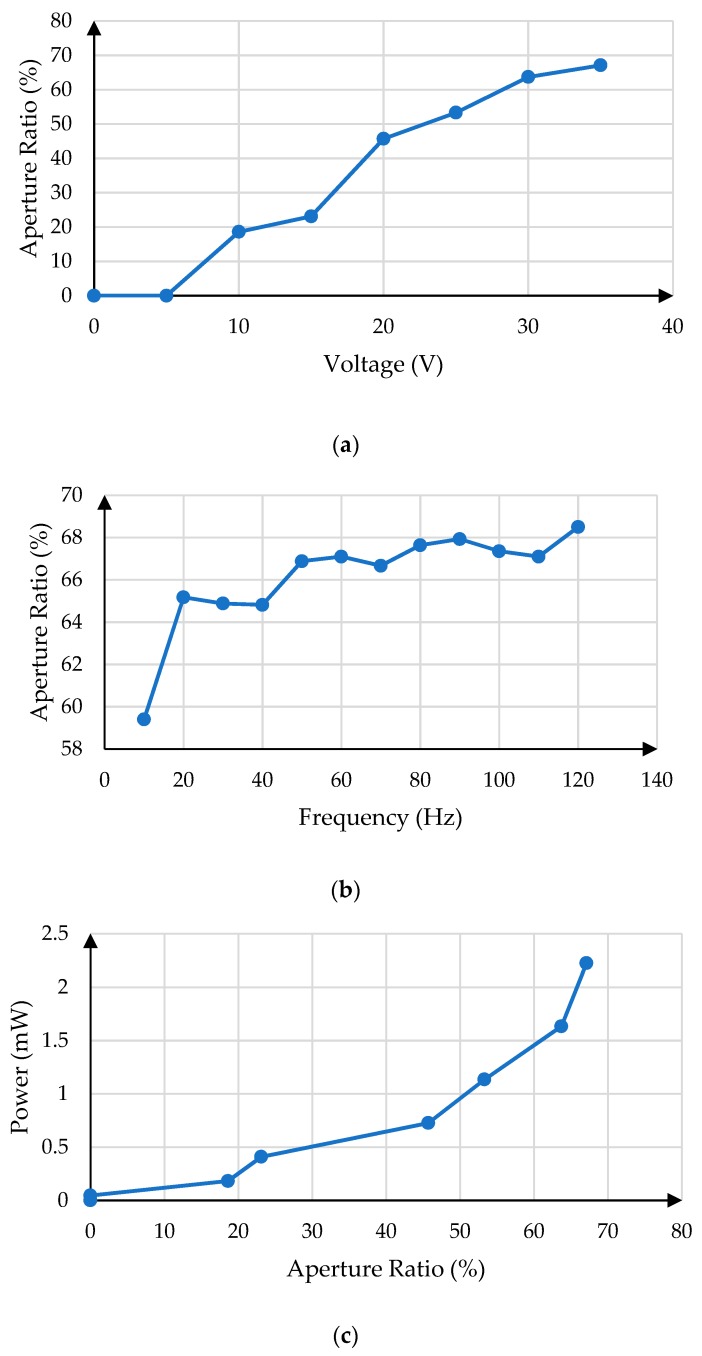
Characteristics of the proposed driving waveform. (**a**) The relationship between voltage and aperture ratio. (**b**) The relationship between frequency and aperture ratio. (**c**) The relationship between aperture ratio and power.

**Table 1 micromachines-11-00145-t001:** Parameters of the electrowetting display (EWD) panel.

Panel Size	Oil Color	Resolution	Pixel Size	Pixel Wall Size	Pixel Wall Height	Hydrophobic Layer Thickness	Driving Voltage
10 × 10 cm	Cyan	200 × 200	150 × 150 um	15 × 15 um	5.6 um	1 um	0–35 V

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
