# Peer review of "Driving Waveform Design with Rising Gradient and Sawtooth Wave of Electrowetting Displays for Ultra-Low Power Consumption"

_micromachines, 2020, doi:10.3390/mi11020145_

Round 1

Reviewer 1 Report

The work reported in the manuscript – hereafter MS – is to optimize the power consumption of electrowetting displays. They report results of a driving waveform scheme that produce of approx. 38% lower power compare to the control. The novelty of this particular study lies on the proposed sawtooth-like waveform. The study has merit for EWDs and can potentially lead to better EWDs’ device design, however I suggest more experimentation and parametrization to reveal what is the effect of the device design to the driving voltage. In general, the work is decently written and presented throughout the MS. Here are my comments that, in my opinion, need to be addressed in the MS:

Could you do a parametric analysis of the sawtooth waveform in terms of amplitude and wavelength? What are the lessons learned from the parametric analysis? What is the turning point in terms of amplitude and wavelength that the system consumption is suboptimal? What is the effect of the hydrophobic layer thickness of the device in the sawtooth waveform? What is the optimal driving waveform If we had, for example, 1.5um thickness or 0.5um thickness? Is it the same with the one presented in the MS or different? What would the sawtooth waveform be in those cases? What is the effect of the pixel wall size and height in the presented waveform? Would this be different if those numbers were different? Evaluation of all above parametrizations with Aperture ratio (%) vs. Power graphs (mW) What is the mechanism(s) that inhibits backflow of the oil in the case of the sawtooth wave? I understand that sawtooth waveform (or PWM in general) may reduce charge trapping (this is known). What is the optimal sawtooth wave and what is the difference with a high freq. square wave for instance instead of sawtooth (or any other high freq. waveform for that matter)? Higher freq. sawtooth with higher amplitude may have lower consumption? Minor suggestions:

Figures 6, 7, 8 & 9 are simple enough that can all be displayed in a single Figure with a, b, c, d

In conclusion, if the aforementioned comments are addressed, I would recommend this MS for publication in Micromachines Journal.

Reviewer 2 Report

Comments to authors

Authors present a paper about Driving Waveform Design with Rising Gradient and Sawtooth Wave of Electrowetting Displays for Ultra-Low Power Consumption
The paper is well referenced and written

Some comments:
1/ Please, give an order of the physical dimensions of the pixel in § 2
2/ C1 C2 are not a well named due to capacitor name conflict (line 124), please adapt your notation to avoid confusion.
3/ Experimental dots should appear in all figures, Fig. 7, 8 and 9
4/ Fig. 8: X-axis legend is not appropriate, please put Rising duration (ms)
5/Fig. 9: explain why the aperture ratio presents fluctuation peaks
6/ Recall the main result in conclusion (as in abstract)

Round 2
